# Sevoflurane versus ketamine+diazepam anesthesia for assessing systemic and hepatic hemodynamics in rats with non-cirrhotic portal hypertension

José Ignacio Fortea[1,2,3], Marta Puerto[1,3], Carolina Fernández-Mena[1], Iris Asensio[1,3], María Arriba[4], Jorge Almagro[1], Juan Bañares[1], Cristina Ripoll[1,3,5], Rafael Bañares[1,3,6], Javier Vaquero[1,3]*

1 HepatoGastro Lab, Servicio de Ap. Digestivo del HGU Gregorio Marañón, Instituto de Investigación Sanitaria Gregorio Marañón (IiSGM), Madrid, Spain, 2 Servicio de Digestivo, Hospital Universitario Marqués de Valdecilla, Santander, Spain, 3 Centro de Investigación Biomédica en Red en Enfermedades Hepáticas y Digestivas (CIBEREHD), Madrid, Spain, 4 Servicio de Bioquímica Clínica del HGU Gregorio Marañón, Instituto de Investigación Sanitaria Gregorio Marañón (IiSGM), Madrid, Spain, 5 Innere Medizin I, Martin-Luther-Universität Halle-Wittenberg, Halle, Germany, 6 Facultad de Medicina, Universidad Complutense, Madrid, Spain

* j.vaquero@iisgm.com

## Abstract

The selection of the anesthetic regime is a crucial component in many experimental animal studies. In rodent models of liver disease, the combination of ketamine and diazepam (KD), generally by the intramuscular (i.m.) route, has traditionally been the anesthesia of choice for the evaluation of systemic and hepatic hemodynamics but it presents several problems. Here, we compared the performance of inhalational sevoflurane (Sevo) against the KD combination as the anesthesia used for hemodynamic studies involving the measurement of portal pressure in normal rats (Ctrl) and rats with non-cirrhotic portal hypertension induced by partial portal vein ligation (PPVL). Compared with Ctrl rats, rats with PPVL presented characteristic alterations that were not influenced by the anesthetic regime, which included liver atrophy, splenomegaly, increased plasma fibrinogen, decreased alkaline phosphatase and glycemia, and frequent ascites. The use of the KD combination presented several disadvantages compared with the inhalational anesthesia with sevoflurane, including considerable mortality, a higher need of dose adjustments to maintain an optimal depth of anesthesia, increases of heart rate, and alteration of blood biochemical parameters such as the concentration of aspartate aminotransferase, lactate, and lactic dehydrogenase. Rats anesthetized with sevoflurane, on the other hand, presented lower respiratory rates. Importantly, the anesthetic regime did not influence the measurement of portal pressure either in Ctrl or PPVL rats, with the increase of portal pressure being similar in Sevo- and KD- anesthetized groups of PPVL rats compared with their respective control groups. Overall, our results suggest that anesthesia with sevoflurane is preferable to the combination of KD for performing systemic and hepatic hemodynamic studies in rats with non-cirrhotic portal hypertension.

**Data Availability Statement:** All relevant data are within the paper and its Supporting Information files.

**Funding:** This work was supported by grants from Instituto de Salud Carlos III-Fondos FEDER (PI15/01083 to JV y PI15/02037 to RB), and from Comunidad de Madrid-Fondos FEDER-FSE (Exohep-CM S2017/BMD-3727 and Nanoliver-CM Y2018/NMT-4949 to RB). The funders had no role in study design, data collection and analysis, decision to publish, or preparation of the manuscript.

**Competing interests:** The authors have declared that no competing interests exist.

## Introduction

Liver cirrhosis—the end-stage of chronic liver diseases—represents a significant healthcare problem with associated mortality that is similar to that of major cancers [1]. Portal hypertension—the increase of portal pressure in the portal vein territory—is responsible for most morbidity and mortality in patients with cirrhosis, constituting the leading cause of death and liver transplantation [2]. Much of our understanding of the pathophysiology of portal hypertension has been generated in experimental models in rats because they allow direct measurements of portal pressure and regional blood flows. Ketamine with or without a benzodiazepine, generally administered intramuscularly (i.m.) instead of intraperitoneally (i.p.) to avoid direct effects of anesthetics on splanchnic vessels, has been the anesthesia of choice in most studies involving measurement of portal pressure in rodents [3–9].

The selection of an appropriate anesthetic is a crucial component of experimental animal research not only from ethical and legal perspectives but also from a methodological point of view [10]. Indeed, inadequate control of pain and distress may interfere with experimental results due to the associated nervous and endocrine stress responses, and the anesthetic itself can also alter the animal model or the parameters that are measured [11]. The species, strain, sex, age, diet, temperature, time of day, frequency and route of administration, or the use of other drugs can also influence the effects of anesthetics [12]. Importantly, most anesthetics are metabolized in the liver, which introduces a relevant additional factor for studies of cirrhosis and portal hypertension as the effects of anesthetics may vary between experimental groups depending on the absence or presence of different degrees of liver dysfunction.

The ideal anesthetic regime for the measurement of portal pressure in experimental studies should balance several universal and disease-specific goals. Specifically, it should: 1) Be precisely titratable to ensure that animals receive adequate anesthesia without experiencing hemodynamic instability or life-threatening overdose, 2) Be compatible with available equipment and medications, 3) Have no hepatotoxicity, 4) Have minimal hepatic metabolism and elimination to limit the impact of liver disease upon its pharmacokinetics, and 5) Have no effects or minor effects on systemic and splanchnic hemodynamic parameters. None of the most used anesthetics in studies of portal hypertension in rodents, however, fulfill these criteria, including the combination of ketamine with benzodiazepines. Indeed, an unpredictable individual sensitivity to ketamine resulting in associated intra-operative mortality of up to 30% has been reported, particularly in cirrhotic rats [13]. A recent study in rats anesthetized with ketamine+-xylazine undergoing diverse surgical operations (median laparotomy, bile-duct ligation, or balloon dilatation of the carotid artery) suggested that hypoxia may be a contributing factor, as oxygen supplementation reduced the mortality [14]. Studies comparing the performance of different anesthetics in the setting of cirrhosis or portal hypertension, however, are scarce and limited to relatively old studies that do not include the newer anesthetic agents [13,15–19] (see Table 1 for a summary).

## Materials and methods

### Animals

Male Sprague-Dawley (SD) rats (310–580 g, n = 41) from Charles River Laboratories (France) were maintained in a 12:12 hour light:dark cycle with constant conditions of room temperature and air humidity. All rats always had free access to standard laboratory food and tap water. A hemodynamic study under anesthesia with inhalational sevoflurane (Sevo) or with ketamine+diazepam (KD) was performed in normal rats (Ctrl) and rats with partial portal

**Table 1. Summary of studies evaluating the effects of anesthesia on hemodynamics in rats with experimental cirrhosis and/or portal hypertension.**

| 1st author Year, Ref | Species/ strain/sex/ BW | Anesthetics doses/route of administration | Main end-points | PP measured | Main results |
|---|---|---|---|---|---|
| Belghiti J, 1981 [16] | Male SD rats<br><br>BW: 200–250 gr | • Control: Awake rats<br><br>• **Halothane** vol % NS | Anesthetic influence on CO and PP in normal and PPVL rats. | YES | • PP was measured four consecutive times in normal and PPVL rats, as follows: anesthetized, awake, re-anesthetized, and re-awakened.<br><br>• Halothane anesthesia decreased PP, MAP, and CO compared to awake rats in normal and PPVL rats. |
| Lee S.S, 1985 [17] | Male SD rats<br><br>BW: NS | • Control: Awake rats<br><br>• **Pentobarbital** sodium: 5 mg/100 mg BW ip | Pentobarbital influence on hemodynamics and blood flow distribution in PPVL and sham-operated rats. | YES | In both Sham-operated and PPVL rats, pentobarbital anesthesia:<br><br>• Decreased CO and HABF, and increased TPR.<br><br>• Did not change absolute blood flow values of splanchnic organs or PBF in either group, but the fractions of CO perfusing splanchnic organs were significantly increased in both groups.<br><br>• PP, HR, and MAP were not influenced by anesthesia in any group. |
| Lee S.S, 1986 [18] | Male SD rats<br><br>BW: NS | • Control: Awake rats<br><br>• **Pentobarbital** sodium: 5 mg/100 mg BW ip. | Pentobarbital influence on hemodynamics and blood flow in BDL and sham-operated rats. | YES | Pentobarbital anesthesia:<br><br>• Decreased CO and increased TPR in both BDL and sham-operated rats.<br><br>• Decreased PBF and HABF, and increased HR and portal resistance in BDL rats.<br><br>• Did not influence PP or MAP in BDL or sham-operated rats.<br><br>**Conclusion**: Pentobarbital anesthesia is not suitable for hemodynamic studies. |
| Debaene B, 1990 [15] | Male SD rats<br><br>BE: 308 ± 5 gr | • Controls: Awake cirrhotic rats<br><br>• **Ketamine**: 30 mg/kg iv bolus followed by a continuous iv infusion of 1.5 mg/kg/min<br><br>• **Halothane** 1%<br><br>• **Enflurane** 2.2%<br><br>• **Isoflurane** 1.3% | Anesthetic influence on systemic hemodynamics and splanchnic blood flow in BDL rats that were normo-or hypo-volemic. | NO | Before hemorrhage:<br><br>• CI was higher in conscious rats and rats receiving isoflurane.<br><br>• No differences in MAP, PBF, or HR between anesthetics.<br><br>• Splanchnic blood flow was lower in enflurane and similar among the rest of anesthetics; but HABF was similar in conscious rats, isoflurane and halothane groups, and lower with ketamine.<br><br>After hemorrhage:<br><br>• CI and PBF decreased similarly in all groups.<br><br>• MAP was significantly higher in the ketamine and isoflurane compared with the enflurane and halothane groups. HABF was similar in conscious rats and isoflurane group.<br><br>All anesthetic agents had the same effect on PBF, but they acted differently on HABF.<br><br>**Conclusion**: Isoflurane seemed the most efficient anesthetic for preserving the splanchnic circulation in hypovolemic cirrhotic rats. |

(Continued)

**Table 1.** (Continued)

| 1st author Year, Ref | Species/ strain/sex/ BW | Anesthetics doses/route of administration | Main end-points | PP measured | Main results |
|---|---|---|---|---|---|
| Sikuler E, 1991 [19] | Male SD rats | • Controls: Awake sham and BDL rats | Influence of ketamine anesthesia on systemic hemodynamics and blood flow distribution in BDL rats. | NOT in awake rats. | Compared with awake sham animals, anesthesia with ketamine: |
| | BW: 220–240 gr | • **Ketamine**: 100–150 mg/kg im | | | • Increased PBF in sham-operated rats, but it did not change MAP, HR, CO, or HABF. |
| | | | | | • Increased MAP in BDL rats, but it did not affect HR, CO, HABF, or PBF. |
| | | | | | • TPR was lower in awake rats compared with anesthetized BDL and sham rats. |
| Van Roey G, 1997 [13] | Male Wistar rats | • **Ether**. | Anesthetic influence on systemic and splanchnic hemodynamics compared with the awake state in normal and cirrhotic rats that were either normo or hypovolemic. | YES | Hemodynamics studies in awake rats were performed 3 hours after recovery from ether narcosis for placement of catheters. |
| | BW: 150 gr | • **Pentobarbital**: 60 and 30 mg/kg ip for normal and cirrhotic rats. | | | • Ether: Dose adjusted to the depth of narcosis. Few hemodynamic effects, but continuous observation of the animal was essential. |
| | | • **Ketamine**: 150 and 75 mg/kg ip for normal and cirrhotic rats. | | | • Pentobarbital: Markedly suppressed the sympathetic nervous system and produced profound hypotension in cirrhotic rats. No influence on PP. |
| | | • **Diacepam-fluanisone**: 2.5–1 mg/kg ip and 2–0.6 mg/kg im for normal-cirrhotic rats. | Cirrhosis was induced by CCl4 administration. | | • Ketamine: Unpredictable individual sensitivity to Ketamine, especially in cirrhotic rats, with a total mortality > 30%. Ketamine did not influence MAP or PP and decreased RR in normal and cirrhotic normovolemic rats. |
| | | | | | • Diazepam-fluanisone: Produced profound hypotension. |
| | | | | | **Conclusion**: Hemodynamic experiments in cirrhotic rats should be preferably performed in awake rats. |

Abbreviations: BDL: common bile duct ligation, BW: body weight, CCl4: carbon tetrachloride, CI: cardiac index, CO: cardiac output, HABF: hepatic arterial blood flow, im: intramuscular, ip: intraperitoneal, iv: intravenous, MAP: mean arterial pressure, NS: not specified, PBF: portal blood flow, PP: portal pressure, PPVL: partial portal vein ligation, RR: respiratory rate, SD: Sprague-Dawley, TPR: total peripheral resistance.

Based on the previous information, the present study aimed to compare the performance of the combination of ketamine+diazepam (KD) given i.m. versus inhalational anesthesia with sevoflurane, which has minimal (< 3%) hepatic metabolism, as the anesthetic regime for measuring hepatic and systemic hemodynamics in rats with non-cirrhotic portal hypertension induced by partial portal vein ligation (PPVL).

vein ligation (PPVL). Therefore, the following four groups of rats were evaluated: Ctrl+Sevo (n = 8), Ctrl+KD (n = 11), PPVL+Sevo (n = 9), and PPVL+KD (n = 13). All studies complied with the Guide for the Care and Use of Laboratory Animals (NIH publication no. 86–23, revised 1985) and with European and local regulations. Ethical approval for the study was obtained from the Comité de Ética en Experimentación Animal of the Hospital General Universitario Gregorio Marañon and the Comunidad de Madrid (PROEX# 272/15).

## Surgery

Pre-hepatic portal hypertension was induced in rats by performing a PPVL, as described by Chojkier et al. [20]. Briefly, the rats underwent a mid-laparotomy under sterile conditions and sevoflurane anesthesia. The omentum and some intestine were gently lifted out of the abdomen and covered with a gauze moistened in physiological saline. After carefully separating the

hepatic artery and the portal vein proximal to the confluence of its right and left branches, a blunt-end 20-gauge needle was placed alongside the portal vein, and a 3–0 silk ligature was tightened around them. The needle was then removed, resulting in a calibrated stenosis of the portal vein. The abdominal viscera were returned into the abdomen, washed with physiological saline, and the abdomen was closed in two layers with 3–0 silk. A subcutaneous dose (0.05 mg/kg) of buprenorphine (0.3 mg/ml Buprenex vials, Reckitt Benckiser) was administered for pain relief and antiseptic ointment was applied to the surgical wound. Body temperature was monitored with a thermometer and maintained at normal levels with a warming pad throughout the operation. Once the animals recovered from the procedure, they were returned to individual cages, and physiological and behavioral signs indicative of pain or discomfort (body position, self-cleaning, body weight, motility, food intake) were assessed daily.

## Anesthetic regimes for the hemodynamic studies

Hemodynamic studies with measurement of portal pressure were performed under one of the following anesthetic protocols:

1. KD: Intramuscular injections of ketamine (75 mg/kg bw, Ketolar®, Pzifer, Madrid, Spain) and diazepam (5 mg/kg bw, Valium®, Roche, Madrid, Spain) were administered separately in the anterolateral side of each thigh. The volume of injection for each anesthetic was between 0.50–0.87 ml for ketamine and 0.33–0.58 ml for diazepam. Additional doses of ketamine (10 mg in 0.2 mL) were administered by the same route when required to maintain a sufficient depth of anesthesia. Rats anesthetized with this protocol breathed room air.

2. Sevoflurane: The equipment for inhalational anesthesia with sevoflurane (Sevorane, Abbvie Spain SLU) consisted of a 100% oxygen tank connected to a flowmeter, a sevoflurane vaporizer (Datum Vaporizer, Blease, England), an induction chamber, and a facemask for the induction and maintenance of anesthesia during the PPVL operation and the hemodynamic study. Induction of anesthesia was performed with 5% sevoflurane in 100% $O_2$, which was reduced to 2–3% sevoflurane for anesthesia maintenance. The percentage of sevoflurane was modified in 0.5% steps if required for adequate narcosis.

Narcosis was judged to be sufficient when animals had no pedal withdrawal reflex and did not react on cutting through the skin or the peritoneal surface. Once the surgery was initiated, the depth of anesthesia was assessed by cardiovascular parameters (heart rate and arterial pressure), breath rate, and the pedal withdrawal reflex.

## Hemodynamic measurements

The hemodynamic study was performed in PPVL rats four days after the operation, as the hyperdynamic splanchnic and systemic circulation characteristic of portal hypertension are well established at this time [6]. No previous fasting was required. After anesthetizing the rats with the corresponding anesthetic regime, the right common carotid artery and external jugular vein were dissected and catheterized using a 24G Abbocath catheter (B. Braun) and a polyethylene tube (PE50, BD Intramedic), respectively, which were immediately connected to a pressure transducer. A mid-laparotomy was then performed, and part of the intestine was gently lifted out of the abdomen and covered with a gauze moistened in physiological saline. A 24G Abbocath catheter (B. Braun) was inserted into the ileocolic vein and connected to a pressure transducer to estimate the portal pressure (PP). After 5–10 minutes of stabilization, the mean arterial pressure (MAP, carotid artery), the central venous pressure (CVP, jugular vein), and the portal pressure were simultaneously registered for 10 minutes and analyzed using a multichannel PowerLab 8/35 and Lab Chart Reader software (AD Instruments). Body

temperature was monitored with a rectal thermometer and always maintained at stable levels (37.0 ± 0.5 ˚C) with a warming pad.

At the end of the hemodynamic studies, the rats had euthanasia by exsanguination, the correct position of the portal ligature was confirmed, and the liver and spleen were dissected and weighted.

## Hematological and biochemical blood laboratory tests

After finalizing the hemodynamic measurements, arterial blood from the carotid artery was collected into tubes containing EDTA or heparin. Blood cell counts were measured by an automated analyzer (BC-2800 Vet Auto Hematology Analyzer, MINDRAY Bio-Medical Electronics Co., Ltd) on EDTA-anticoagulated blood specimens. Plasma from heparinized blood was used to measure the concentration of glucose, fibrinogen, alanine (ALT) and aspartate (AST) aminotransferases, alkaline phosphatase (AP), lactic dehydrogenase (LDH), bilirubin, and albumin in an automated analyzer (ADVIA Chemistry XPT System, Siemens). Arterial blood gases were measured in a GEM Premier 3000 analyzer (Instrumentation Laboratory, Werfen).

## Statistical analysis

Quantitative variables were expressed as median [interquartile range] and qualitative variables as proportions (%). Comparisons between two groups were performed using the Mann-Whitney U test. Comparisons between three or more groups were performed using One-way ANOVA with Tukey's post-hoc tests, or the Kruskal-Wallis test with Dunn's post-hoc tests if there was no equality of variances. Equality of variance was analyzed using the Brown-Forsythe test. Proportions were compared using Fisher's exact test. Raw data of the experiments are provided as Supporting Information (S1 File). Statistical analyses were performed using GraphPad Prism v8 (GraphPad Software LLC, San Diego, CA).

## Results

### General characteristics

Rats in the PPVL groups showed the expected changes after the calibrated stenosis of the portal vein, including decreased liver-to-body weight (BW) ratio (Ctrl: 3.41% [3.16, 3.72] vs. PPVL: 2.68% [2.58, 2.93], p< 0.0001), increased spleen-to-BW ratio (Ctrl: 0.192% [0.184, 0.205] vs. PPVL: 0.265% [0.219, 0.299], p< 0.0001), and frequent development of ascites (Ctrl: 0/15 (0%) vs. PPVL: 11/21 (52.4%), p = 0.0007) (Fig 1A–1C). Ascites in PPVL rats was generally minimal (less than 1.5 ml), although one rat of each group showed higher values (8 ml and 2 ml, in the PPVL+Sevo and the PPVL+KD groups, respectively). There were no differences in any of these parameters between PPVL+Sevo and PPVL+KD groups.

BW loss from baseline values before the PPVL surgery to the day of the hemodynamic study was similar in PPVL+Sevo and PPVL+KD rats (-3.8% [-6.63, -0.47] vs. -5.53% [-7.36, -3.08], p = 0.35). However, all groups showed similar BW at the time of the hemodynamic study (p = 0.7, Table 2).

### Anesthesia-related mortality with the KD combination

Mortality only occurred in rats anesthetized with KD (Sevo: 0/17 (0%) vs. KD: 5/24 (20.8%), p = 0.065, Fig 2A), with rats in the Ctrl+KD group showing the highest mortality compared to the other groups (4/11 (36.4%) vs. 1/30 (3.3%), p = 0.014). All deaths occurred during the hemodynamic measurements several minutes after the cannulation of blood vessels. All dying

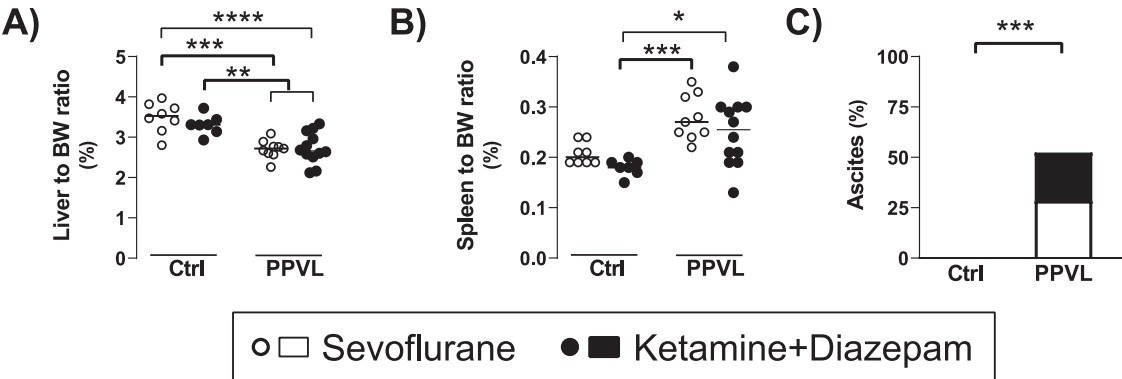

**Fig 1. A) Liver to body weight (BW) ratio, B) spleen to BW ratio, and C) proportion of rats presenting ascites in normal rats (Ctrl) and rats with partial portal vein ligation (PPVL) anesthetized with sevoflurane (white dots/bar portion) or ketamine +diazepam (black dots/bar portion).** $*$ p< 0.05, $**$ p< 0.01, $***$ p< 0.001, p< 0.0001 ANOVA with Tukey's post-hoc test (A), Kruskall-Wallis with Dunn's post-hoc tests (B), or Fisher's exact test (C). The line and dots represent the median and individual rat values, respectively.

rats presented progressive hypotension without any detectable hemorrhage, respiratory depression, or signs of inadequate depth of anesthesia.

## Stability of depth of anesthesia and need of additional doses of anesthetics

Adequate depth of anesthesia was achieved regardless of the anesthetic choice, but the rats anesthetized with KD required more frequent dose adjustments (Sevo: 3/17 (18%) vs. KD: 12/ 24 (50%), p = 0.050, Fig 2B). Additional administrations of ketamine (10 mg in 0.2 ml) to maintain an optimal depth of anesthesia were required once in three rats of the Ctrl+KD group and six rats of the PPVL+KD group, and twice in three additional rats of the Ctrl+KD group. The duration of the hemodynamic study was similar in all the groups (p = 0.15, Table 2).

## Influence of anesthetic regimes on cardio-respiratory parameters

The anesthetic choice had an impact on respiratory and heart rates. Specifically, rats anesthetized with KD presented an increase of heart rate and rats anesthetized with sevoflurane presented a decrease of respiratory rate with respect to the rates expected in the awake state [21] (Fig 3A and 3B). All groups showed similar MAP and CVP regardless of the type of anesthetic or the PPVL surgery (Fig 3C and 3D).

**Table 2. Physiological, procedural and blood biochemical parameters of rats at the time of the hemodynamic study.**

|  | Ctrl+Sevo | Ctrl+KD | PPVL+Sevo | PPVL+KD | p |
|---|---|---|---|---|---|
| **Body weight (g)** | 390 [360, 412] | 393 [388, 402] | 375 [341, 396] | 365 [335, 421] | 0.7 |
| **Duration of hemodynamic study (min)** | 25.3 [21.6, 27.4] | 22.5 [21.0, 25.0] | 21.0 [21.0, 23.5] | 21.0 [21.0, 25.0] | 0.15 |
| **ALT (U/l)** | 26 [23, 28] | 35 [30, 37] | 28 [22, 35] | 34 [28, 50] | 0.18 |
| **Bilirubin (mg/ml)** | 0.1 [0.1, 0.1] | 0.1 [0.05, 0.1] | 0.1 [0.1, 0.2] | 0.1 [0.1, 0.1] | 0.19 |
| **Albumin (g/dl)** | 3.1 [3.1, 3.4] | 3.0 [2.6, 3.5] | 2.7 [2.6, 3.0] | 3.1 [2.5, 3.2] | 0.38 |

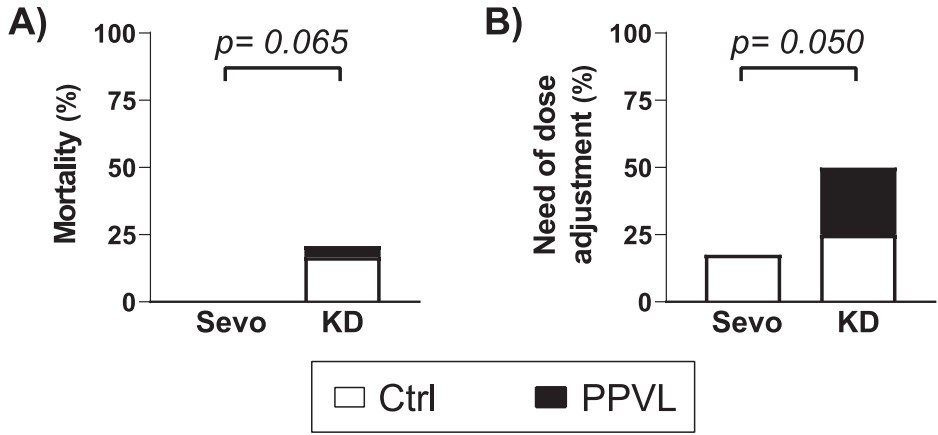

**Fig 2. A) Mortality (%) and B) proportion of rats requiring adjustment of the dose in normal rats (Ctrl, white bar portion) and rats with partial portal vein ligation (PPVL, black bar portion) anesthetized with sevoflurane (Sevo) or with ketamine+diazepam (KD).** Comparisons were performed with Fisher's exact test.

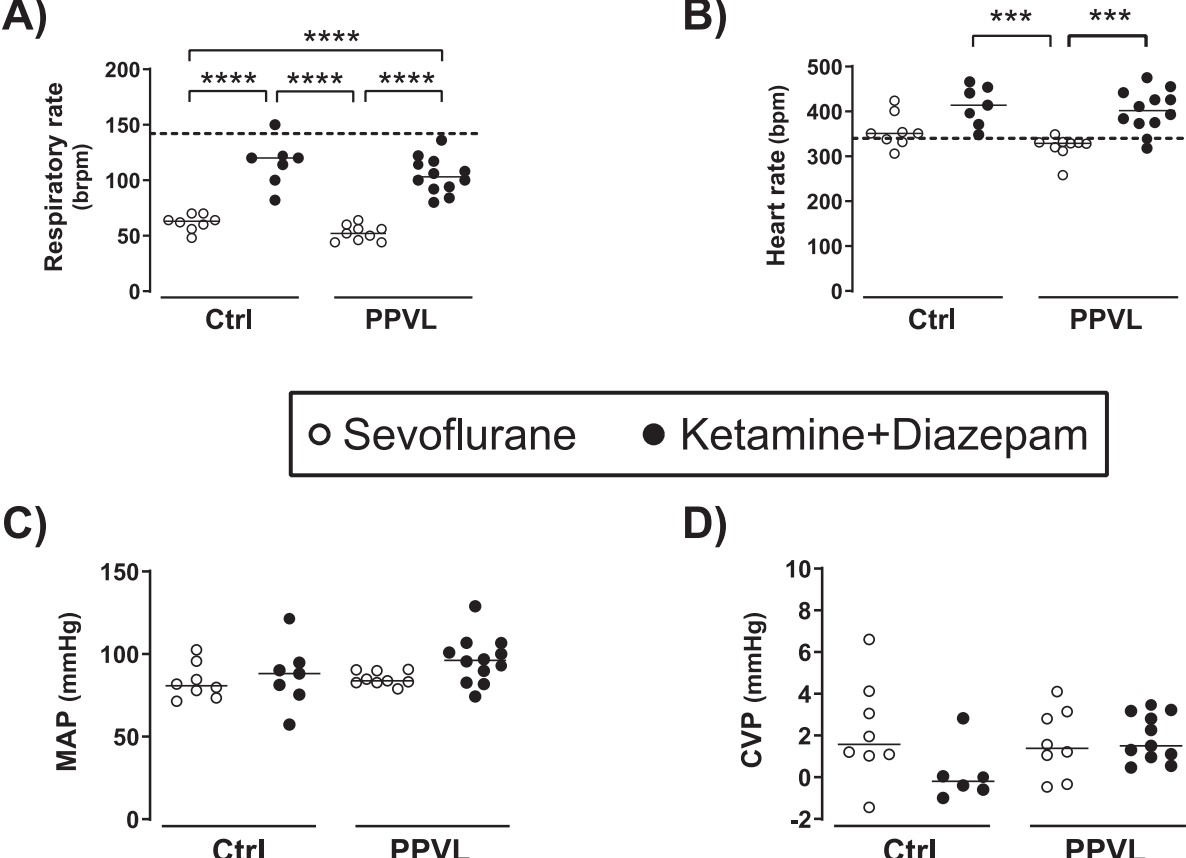

**Fig 3. A) Respiratory rate, B) heart rate, C) mean arterial pressure (MAP), and D) central venous pressure (CVP) in normal rats (Ctrl) and rats with partial portal vein ligation (PPVL) anesthetized with sevoflurane (Sevo) or ketamine+diazepam (KD).** Median [IQR] of Ctrl+Sevo vs. Ctrl+KD vs. PPVL+Sevo vs. PPVL+KD for: *Respiratory rate (brpm)*: 63 [57, 69] vs. 120 [100, 122] vs. 52 [45, 58] vs. 103 [93, 116]; *Heart rate (bpm)*: 351 [334, 389] vs. 414 [371, 454] vs. 329 [316, 332] vs. 402 [374, 438]; *MAP (mmHg)*: 81 [75, 93] vs. 88 [76, 95] vs. 84 [83, 90] vs. 96 [85, 105]; *CVP (mmHg)*: 1.6 [1.0, 3.9] vs. -0.2 [-0.7, 0.7] vs. 1.4 [0.0, 3.1] vs. 1.5 [1.0, 3.2]. *** $p < 0.001$, **** $p < 0.0001$ One-way ANOVA with Tukey's post-hoc tests. The line and dots represent the median and individual rat values, respectively. The dash lines represent the mean respiratory rate (142 brpm) and heart rate (340 bpm) reported in normal awake rats in prior studies [21].

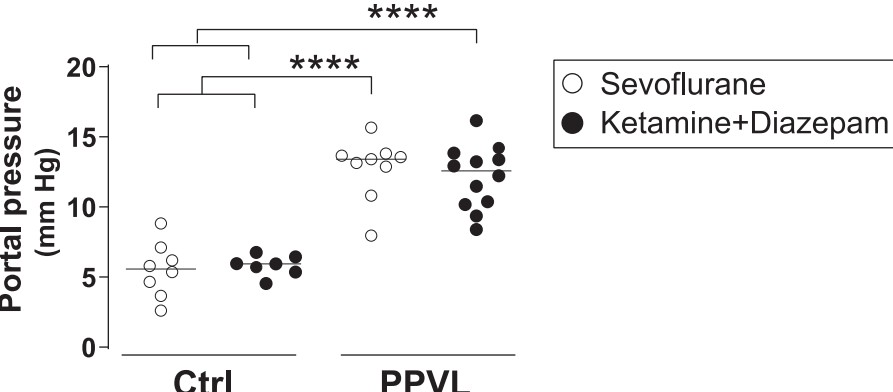

**Fig 4. Portal pressure (mmHg) in normal rats (Ctrl) and rats with partial portal vein ligation (PPVL) anesthetized with sevoflurane (Sevo) or ketamine+diazepam (KD).** **** p< 0.0001 One-way ANOVA with Tukey's post-hoc tests. The line and dots represent the median and individual rat values, respectively.

## The anesthetic choice did not influence the measurement of portal pressure

Compared with Ctrl animals, rats undergoing PPVL presented an increase in portal pressure (p< 0.0001) that was similar in sevoflurane- and KD-anesthetized rats (Ctrl+Sevo: 5.6 mmHg [3.9, 6.9] vs. Ctrl+KD: 5.9 mmHg [5.4, 6.4] vs. PPVL+Sevo: 13.4 mmHg [11.8, 13.7] vs. PPVL +KD: 12.6 mmHg [10.2, 13.7], p< 0.0001 PPVL groups vs. Ctrl groups, Fig 4). Importantly, there were no differences in portal pressure between sevoflurane- and KD-anesthetized Ctrl groups.

## Anesthesia- and PPVL-related alterations in blood biochemistry

Changes in blood biochemical parameters were related to the PPVL surgery as well as to the choice of anesthetic. Compared with Ctrl rats, both sevoflurane- and KD-anesthetized PPVL rats presented elevation of fibrinogen and slight decreases of plasma glucose and alkaline phosphatase (Fig 5A–5C). Compared with rats anesthetized with sevoflurane, both Ctrl and PPVL rats anesthetized with KD presented increases of AST, lactate, and LDH (Fig 5D–5F). All groups (Ctrl+Sevo, Ctrl+KD, PPVL+Sevo, PPVL+KD) showed normal values of ALT (p = 0.18), bilirubin (p = 0.19), and albumin (p = 0.38) (Table 2).

There were also differences in the arterial blood gases at the end of the hemodynamic study between sevoflurane- and KD- anesthetized rats. Thus, rats in the groups anesthetized with sevoflurane presented higher pH (7.47 [7.44, 7.48] vs. 7.39 [7.37, 7.43], p = 0.0014), partial pressure of $O_2$ (462 mmHg [417, 483] vs. 81 mmHg [77, 89], p< 0.0001), partial pressure of $CO_2$ (37 mmHg [34, 44] vs. 29 mmHg [24, 38], p = 0.0198), bicarbonate (26.9 mM [25.0, 28.9] vs. 19.1 mM[13.7, 24.2], p = 0.0002), and $O_2$ saturation (100% [100, 100] vs. 96% [95, 97], p< 0.0001) compared with the groups anesthetized with KD.

## Discussion

The use of anesthesia is a crucial component of research involving surgical procedures in experimental animal models, both from an ethical perspective as well as to prevent potential distress-induced interferences on parameters under study [12]. In the case of hemodynamic studies in rats with liver disease and portal hypertension, the combination of ketamine with benzodiazepines (diazepam or midazolam) has traditionally been the anesthetic regime of

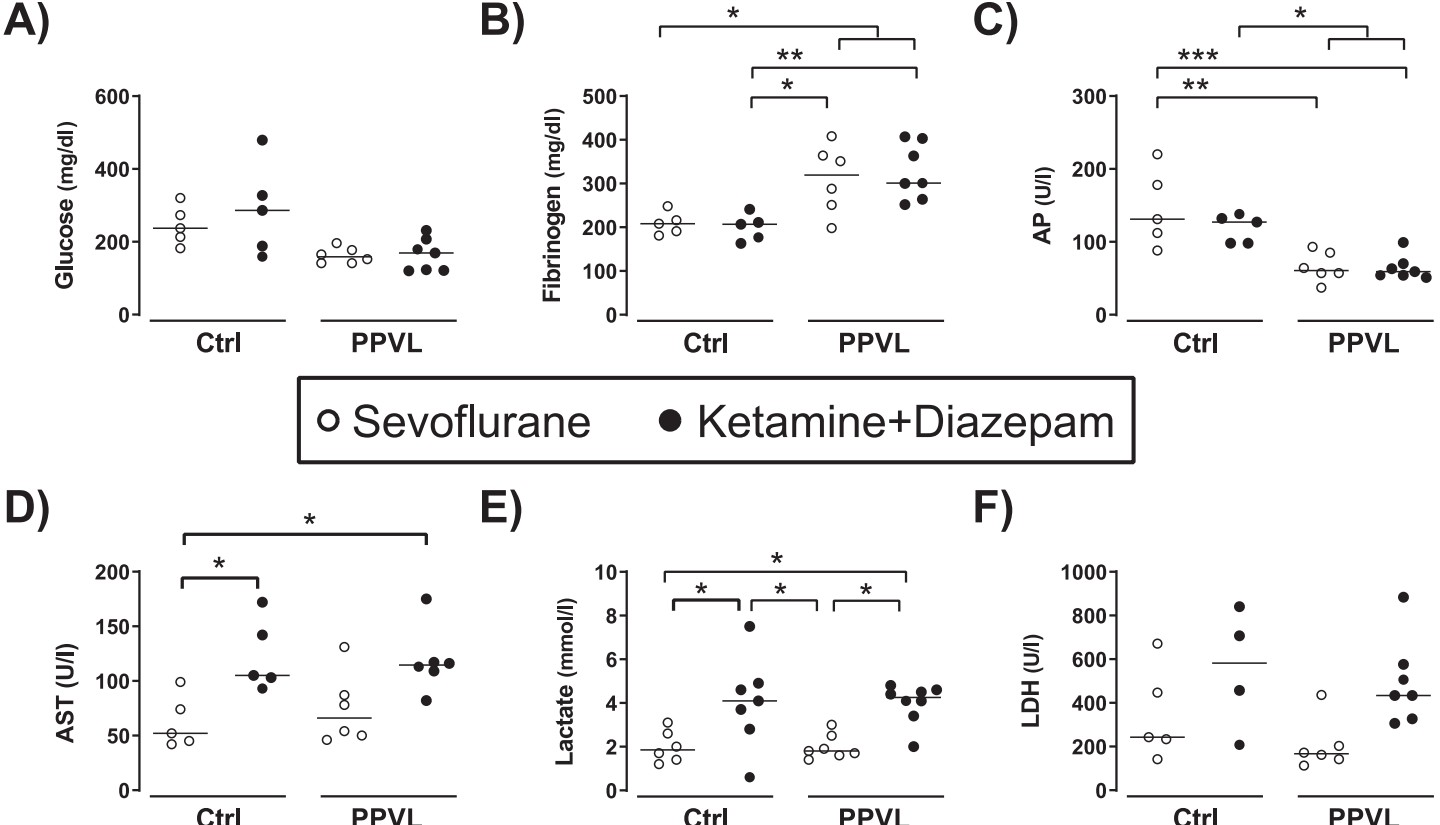

**Fig 5. A) Glucose, B) fibrinogen, C) alkaline phosphatase (AP), D) aspartate aminotransferase (AST), E) lactate, and F) lactic dehydrogenase (LDH) in normal rats (Ctrl) and rats with partial portal vein ligation (PPVL) anesthetized with sevoflurane (Sevo) or ketamine+diazepam (KD).** Median [IQR] of Ctrl+Sevo vs. Ctrl+KD vs. PPVL+Sevo vs. PPVL+KD for: *Glucose (mg/dl)*: 237 [198, 297] vs. 286 [174, 403] vs. 159 [141, 182] vs. 169 [121, 207]; *Fibrinogen (mg/dl)*: 208 [186, 232] vs. 207 [170, 226] vs. 320 [238, 375] vs. 301 [264, 403]; *AP (U/l)*: 131 [100, 199] vs. 127 [98, 135] vs. 61 [52, 87] vs. 59 [54, 70]; *AST (U/ml)*: 52 [44, 87] vs. 105 [98, 157] vs. 66 [49, 98] vs. 115 [102, 132]; *Lactate (mmol/l)*: 1.9 [1.4, 2.7] vs. 4.1 [2.8, 4.9] vs. 1.8 [1.6, 2.5] vs. 4.3 [3.6, 4.6]; *LDH (U/l)*: 243 [188, 559] vs. 582 [270, 807] vs. 167 [135, 261] vs. 434 [327, 576]. * $p < 0.05$, ** $p < 0.01$, *** $p < 0.001$, **** $p < 0.0001$ One-way ANOVA with Tukey's post-hoc tests. The line and dots represent the median and individual rat values, respectively.

choice [3–9]. This combination, however, is not ideal, as these agents are mainly metabolized in the liver, and it associates potential cardiovascular effects and considerable mortality. The results of the present study indicate that inhalational anesthesia with sevoflurane is preferable to the combination of i.m. KD for assessing hepatic and systemic hemodynamics in rats, as sevoflurane circumvented many of the pitfalls related to the KD combination without interfering with the measurement of portal pressure.

The main observation of our study was that both KD and sevoflurane allowed a correct measurement of portal pressure and detection of portal hypertension in rats with PPVL. Although no measurement of splanchnic flow was performed, the similar increase of portal pressure in PPVL rats suggests that both anesthetic regimes had a similar effect (or no effect) on the hyperdynamic splanchnic and systemic circulation induced by the PPVL, as pre-hepatic resistance was "fixed" by the PPVL operation. Other features induced by PPVL in rats were also similar in both anesthetic groups, including the degree of liver atrophy and splenomegaly, the increase of fibrinogen or the decrease of alkaline phosphatase in plasma, a trend to lower glycemia, and the proportion of rats presenting mild ascites. Together, these observations support that the experimental model of PPVL in rats was well performed and that both anesthetic

regimes were equally effective for the assessment of portal pressure in rats with non-cirrhotic portal hypertension.

Another important finding of the present study was that anesthesia with sevoflurane overcame relevant shortcomings related to the KD combination. First, sevoflurane entirely prevented anesthesia-related mortality as opposed to the 21% mortality observed in the group anesthetized with KD. Of note, KD-related mortality in our study compares favorably against other prior studies, in which up to 30–60% intra- and peri-operative mortality rates have been reported [13,14].

Second, sevoflurane allowed easier maintenance of adequate narcosis, which contrasted with the frequent need for additional doses in rats anesthetized with KD. Indeed, the unpredictable individual sensitivity to ketamine is considered the major problem with this anesthetic agent. Noteworthy, this problem is likely to be further magnified in rats with liver disease, as cytochrome P450 enzymes extensively metabolize ketamine in the liver [22]. Although its combination with diazepam improves the depth of anesthesia and prevents some undesirable effects such as muscle hypertonicity or hypersalivation, interference between ketamine and diazepam at the level of cytochrome P450 enzymes in the liver may also occur [23]. In contrast to KD, less than 5% of sevoflurane is metabolized by the liver [24], and it is a recommended anesthetic in patients with liver disease [25]. It is important to point out, however, that the PPVL model induces pre-hepatic portal hypertension, which limits the extrapolation of our findings to animal models of cirrhosis (e.g. carbon tetrachloride). Future studies would be needed to assess if the performance of the KD anesthetic regime is altered further by the potential pharmacokinetic and pharmacodynamic derangements induced by cirrhosis. Because of the pharmacokinetics and mode of administration (continuous vs. individual boluses), sevoflurane is likely to provide more stable and reproducible levels of narcosis during the registration of hemodynamic parameters than the KD combination, a possibility that was supported by the results of our study.

Third, all rats anesthetized with the KD combination presented alterations of biochemical parameters in plasma that were not present in sevoflurane-anesthetized animals. In particular, rats anesthetized with KD showed elevations of AST, lactate, and LDH, which are relevant parameters for assessing liver damage. Although our study cannot conclusively establish the exact cause of these alterations, we think that muscle cell damage secondary to the i.m. injection of KD is the most likely explanation, as the ALT was normal and the volumes needed to administer the corresponding doses of KD largely exceeded those recommended for the i.m. route (between 150% to 435% higher) [26]. Other alternatives explanations are also possible, such as the increased susceptibility to hypoxia suggested in a recent study [14]. In this regard, significant differences in arterial blood gases parameters were present between sevoflurane- and KD- anesthetized animals that survived the full study. Despite such differences, the alteration of most parameters of arterial blood gases in KD-anesthetized rats was relatively minor, and all of them presented $O_2$ saturations above 92%, making it unlikely that hypoxia was entirely responsible for the biochemical alterations.

Finally, both control and PPVL rats anesthetized with KD presented higher heart rates than those reported in prior studies for normal awake SD rats [21]. Such an increase of heart rate by the KD combination, which has also been reported in other hemodynamic studies in SD rats [27], could affect the evaluation of cardiac output or other hemodynamic parameters and should be considered when interpreting these measurements. On the other hand, anesthesia with sevoflurane did not appear to affect heart rates, but it induced a significant decrease in the respiratory rate that was similar in control and PPVL rats. Neither sevoflurane nor the KD combination appeared to affect MAP or CVP.

Although it may be favorable from an experimental point of view, the use of sevoflurane presents some logistical disadvantages compared with the portability and minimal equipment required for the KD combination. Noteworthy, sevoflurane is expensive and requires relatively complex and bulky equipment for ensuring a correct administration to the animal as well as to prevent undesired health hazards to laboratory personnel. The increased cost of sevoflurane and its equipment, however, should be contrasted against the cost of loss of animals and failed experiments associated with the KD combination.

In conclusion, the choice of anesthetic regimes should balance practical considerations, animal welfare issues, and the potential interference with targeted research parameters. Although both sevoflurane and the KD combination allowed the appropriate measurement of portal pressure, our results indicate that sevoflurane anesthesia is a preferable option for those laboratories with access to equipment for inhalational anesthesia.

## Supporting information

**S1 File. Database with raw data of the experimental study.**
(XLSX)

## Acknowledgments

We would like to thank the personnel of the animal and surgical facilities of the Unidad de Medicina y Cirugía Experimental of the Instituto de Investigación Sanitaria Gregorio Marañón for their help with the performance of this study.

## Author Contributions

**Conceptualization:** Javier Vaquero.

**Data curation:** José Ignacio Fortea, Javier Vaquero.

**Formal analysis:** José Ignacio Fortea, Javier Vaquero.

**Funding acquisition:** Cristina Ripoll, Rafael Bañares, Javier Vaquero.

**Investigation:** José Ignacio Fortea, Marta Puerto, Carolina Fernández-Mena, Iris Asensio, María Arriba, Jorge Almagro, Juan Bañares, Javier Vaquero.

**Project administration:** Javier Vaquero.

**Supervision:** Rafael Bañares, Javier Vaquero.

**Validation:** José Ignacio Fortea, Javier Vaquero.

**Visualization:** José Ignacio Fortea, Javier Vaquero.

**Writing – original draft:** José Ignacio Fortea.

**Writing – review & editing:** José Ignacio Fortea, Marta Puerto, Carolina Fernández-Mena, Iris Asensio, Cristina Ripoll, Rafael Bañares, Javier Vaquero.

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
