## [Decision Letter · Decision Letter 0]

14 Apr 2020

PONE-D-20-08262

Sevoflurane versus ketamine+diazepam anesthesia for assessing systemic and hepatic hemodynamics in rats with portal hypertension.

PLOS ONE

Dear Dr Vaquero,

Thank you for submitting your manuscript to PLOS ONE. After careful consideration, we feel that it has merit but does not fully meet PLOS ONE’s publication criteria as it currently stands. Therefore, we invite you to submit a revised version of the manuscript that addresses the points raised during the review process.

We would appreciate receiving your revised manuscript by May 29 2020 11:59PM. To enhance the reproducibility of your results, we recommend that if applicable you deposit your laboratory protocols in protocols.io, where a protocol can be assigned its own identifier (DOI) such that it can be cited independently in the future. For instructions see: http://journals.plos.org/plosone/s/submission-guidelines#loc-laboratory-protocols

We look forward to receiving your revised manuscript.

Kind regards,

Matias A Avila, Ph.D.

Academic Editor

PLOS ONE

Journal Requirements:

2. Thank you for including your ethics statement:  "All studies complied with the Guide for the Care and Use of Laboratory Animals (NIH publication no. 86-23, revised 1985) and with local regulations (PROEX # 272/15)."

a.) Please amend your current ethics statement to include the full name of the ethics committee that approved your specific study.

b.) Please amend your current ethics statement to confirm that your named ethics committee specifically approved this study.

For additional information about PLOS ONE submissions requirements for ethics oversight of animal work, please refer to http://journals.plos.org/plosone/s/submission-guidelines#loc-animal-research  

3. At this time, we request that you  please report additional details in your Methods section regarding animal care, as per our editorial guidelines:  

(1) Please state the source and number of mice used in the study  

(2) Please describe the post-operative care received by the animals, including the frequency of monitoring and the criteria used to assess animal health and well-being.

Thank you for your attention to these requests.

Reviewers' comments:

Reviewer's Responses to Questions

**Comments to the Author**

1. Is the manuscript technically sound, and do the data support the conclusions?

Reviewer #1: Yes

Reviewer #2: Yes

2. Has the statistical analysis been performed appropriately and rigorously? 

Reviewer #1: Yes

Reviewer #2: Yes

3. Have the authors made all data underlying the findings in their manuscript fully available?

Reviewer #1: Yes

Reviewer #2: Yes

4. Is the manuscript presented in an intelligible fashion and written in standard English?

Reviewer #1: Yes

Reviewer #2: Yes

5. Review Comments to the Author

Reviewer #1: Fortea and colleagues compared the performance of two anesthetic options in a pre-clinical model of non-cirrhotic portal hypertension, and in control animals. Authors demonstrate that ketamine+diazepam has several disadvantages including high mortality and modification of biochemical tests, which were not observed in the sevoflurane group. The topic of the study is relevant, especially considering the diverse anesthetic options and administration routes that are being used in the pre-clinical scenario. Thus, the study is interesting and, also, elegantly developed. The researchers have deep experience in the field of clinical and pre-clinical portal hypertension. I only have minor comments.

- Methods. Number of animals per group is not homogeneous. Could authors comment on the possible mortality due to PPVL procedure or other reasons leading to such variance.

- Results. Portal pressure measurement. Authors observe no differences in PP values when comparing both groups of PPVL animals. This could be predicted considering that in the PPVL model a major contributor to the increment in the porto-vascular resistance (and therefore in PP) is the physical knot that indeed modifies the pre-hepatic resistance. Splanchnic hemodynamic measurement would be interesting to further assess the effects of both anesthetics. Could authors comment here and in the manuscript?

- Results. How authors interpret their findings on sevoflurane-associated mortality? Why was it higher in the ctrl vs the ppvl group?

- Discussion. Although this is an interesting and timely study, these same analyses would be desirable in pre-clinical models of cirrhotic portal hypertension where hepatic function is compromised and, probably, hepatic metabolism of anesthetics would play a bigger role. This limitation/future opportunity should be discussed in the manuscript. Also, I would suggest to include “non-cirrhotic portal hypertension” in the title and abstract of the manuscript.

Reviewer #2: The authors have contributed with a very nice piece of work on the method of anesthesia in experiments models of non-cirrhotic portal hypertension.

the data are convincing and performed state of the art, yet I have some minor suggestions:

1. the title and through out the manuscript the authors should state that these are findings on non-cirrhotic portal hypertension. they cannot generalize those findings in cirrhotic models such as BDL and CCL4, this should be seen at first sight.

2. the data presented as text, should be presented in tables, e.g. BW, duration of the measurements, biochemistry and blood-gas-analysis.

3. in figure 1 A and B the two different groups of each sevoflurane and ketamine/diazepam should be plotted besides each other not together.

6. PLOS authors have the option to publish the peer review history of their article (what does this mean?). If published, this will include your full peer review and any attached files.

Reviewer #1: No

Reviewer #2: No

---

## [Author Response · Author response to Decision Letter 0]

7 May 2020

Reviewer #1: 

Methods

1. Number of animals per group is not homogeneous. Could authors comment on the possible mortality due to PPVL procedure or other reasons leading to such variance.

No mortality due to the PPVL procedure was observed. The variability in the number of animals between groups was due to the increased mortality in the groups anesthetized with Ketamine-Diazepam (KD) for the hemodynamic study. In order to have similar group sizes with available hemodynamic data, we included more rats in both KD groups.

Results

2. Portal pressure measurement. Authors observe no differences in PP values when comparing both groups of PPVL animals. This could be predicted considering that in the PPVL model a major contributor to the increment in the porto-vascular resistance (and therefore in PP) is the physical knot that indeed modifies the pre-hepatic resistance. Splanchnic hemodynamic measurement would be interesting to further assess the effects of both anesthetics. Could authors comment here and in the manuscript?

We agree with Reviewer 1 that a major contributor to the increase in PP in the PPVL model is the increase in the pre-hepatic resistance caused by the physical knot. Although anesthesia cannot affect this “fixed resistance”, four days after the PPVL operation (time at which the hemodynamic study was performed) the hyperdynamic splanchnic and systemic circulation characteristic of portal hypertension are well established and contribute to the increase in PP (Sikuler et al, Am J Physiol. 1985;248: G618-625). The anesthetic regime may influence these other factors, and therefore affect PP values. Measurement of splanchnic flow as suggested by the reviewer would have provided valuable data concerning this issue.

We have added a comment in the 2nd paragraph of Discussion (page 20): “Although no measurement of splanchnic flow was performed, the similar increase of portal pressure in PPVL rats suggests that both anesthetic regimes had a similar effect (or no effect) on the hyperdynamic splanchnic and systemic circulation induced by the PPVL, as pre-hepatic resistance was “fixed” by the PPVL operation.”.

3. How authors interpret their findings on sevoflurane-associated mortality? Why was it higher in the ctrl vs the PPVL group?

No mortality was observed in control and PPVL groups anesthetized with sevoflurane. As far as the higher KD-associated mortality found in the Ctrl vs the PPVL group is concerned, we cannot provide a solid explanation. We can speculate that it may be partially related to the unpredictable individual sensitivity to ketamine, which is unlikely to be influenced by the hemodynamic changes induced by the PPVL procedure. To confirm this hypothesis, however, would likely require very large sample-size of both groups.

Discussion

4. Although this is an interesting and timely study, these same analyses would be desirable in pre-clinical models of cirrhotic portal hypertension where hepatic function is compromised and, probably, hepatic metabolism of anesthetics would play a bigger role. This limitation/future opportunity should be discussed in the manuscript. Also, I would suggest to include “non-cirrhotic portal hypertension” in the title and abstract of the manuscript.

We agree with both reviewers in this issue. This limitation has been included in the 4th paragraph of Discussion (page 21): 

“It is important to point out, however, that the PPVL model induces pre-hepatic portal hypertension, which limits the extrapolation of our findings to animal models of cirrhosis (e.g. carbon tetrachloride). Future studies would be needed to assess if the performance of the KD anesthetic regime is altered further by the potential pharmacokinetic and pharmacodynamic derangements induced by cirrhosis”.

We have also included “non-cirrhotic portal hypertension” in the title, abstract and throughout the manuscript.

Reviewer #2:

1. The title and throughout the manuscript the authors should state that these are findings on non-cirrhotic portal hypertension, they cannot generalize those findings in cirrhotic models such as BDL and CCL4, this should be seen at first sight.

We agree with both reviewers in this issue. This limitation has been included in the 4th paragraph of Discussion (page 21): 

“It is important to point out, however, that the PPVL model induces pre-hepatic portal hypertension, which limits the extrapolation of our findings to animal models of cirrhosis (e.g. carbon tetrachloride). Future studies would be needed to assess if the performance of the KD anesthetic regime is altered further by the potential pharmacokinetic and pharmacodynamic derangements induced by cirrhosis”.

We have also included “non-cirrhotic portal hypertension” in the title, abstract and throughout the manuscript.

2. The data presented as text, should be presented in tables, e.g. BW, duration of the measurements, biochemistry, and blood-gas-analysis.

A new Table has been added with the data of BW, duration of the hemodynamic study and blood biochemistry. as suggested by the reviewer (Table 2, page 15). We preferred to leave arterial blood gas data in the text, as we presented compounded data of the Ctrl and PPVL groups of each anesthetic group.

3. In Figure 1A and B the two different groups of each sevoflurane and Ketamine/Diazepam should be plotted besides each other not together.

Figure 1 has been modified according to these recommendations.

---

## [Editor Report · Decision Letter 1]

13 May 2020

Sevoflurane versus ketamine+diazepam anesthesia for assessing systemic and hepatic hemodynamics in rats with non-cirrhotic portal hypertension.

PONE-D-20-08262R1

Dear Dr. Vaquero,

We are pleased to inform you that your manuscript has been judged scientifically suitable for publication and will be formally accepted for publication once it complies with all outstanding technical requirements.

With kind regards,

Matias A Avila, Ph.D.

Academic Editor

PLOS ONE
---

## [Editor Report · Acceptance letter]

19 May 2020

PONE-D-20-08262R1 

Sevoflurane versus ketamine+diazepam anesthesia for assessing systemic and hepatic hemodynamics in rats with non-cirrhotic portal hypertension. 

Dear Dr. Vaquero:

I am pleased to inform you that your manuscript has been deemed suitable for publication in PLOS ONE. Congratulations! Your manuscript is now with our production department. 

With kind regards,

on behalf of

Dr Matias A Avila 

Academic Editor

PLOS ONE